# Psychometric Testing and Validation of the Italian Version of the Helsinki Chronic Pain Index (I-HCPI) in Dogs with Pain Related to Osteoarthritis

**DOI:** 10.3390/ani14010083

**Published:** 2023-12-25

**Authors:** Giorgia della Rocca, Carlo Schievano, Alessandra Di Salvo, Anna K. Hielm-Björkman, Maria Federica della Valle

**Affiliations:** 1Research Center on Animal Pain, Department of Veterinary Medicine, University of Perugia, 06126 Perugia, Italy; alessandra.disalvo@unipg.it; 2Innovative Statistical Research SRL, 35123 Padova, Italy; cs@i-stat.it; 3Department of Equine and Small Animal Medicine, Faculty of Veterinary Medicine, University of Helsinki, P.O. Box 66, 00014 Helsinki, Finland; anna.hielm-bjorkman@helsinki.fi; 4CeDIS (Centro di Documentazione e Informazione Scientifica), Innovet Italia SRL, 35030 Saccolongo, Italy; fb@innovet.it

**Keywords:** Helsinki Chronic Pain Index, chronic pain, dog, osteoarthritis, Italian validation

## Abstract

**Simple Summary:**

Like non-verbal humans, animals cannot self-report their pain, which needs to be estimated by a proxy. To limit subjective interpretation, specific questionnaires are entering the clinical practice. The present study aimed to develop and validate the Italian version of an owner-completed questionnaire, originally created in Finnish, for assessing chronic pain in dogs suffering from osteoarthritis, i.e., the so-called Helsinki Chronic Pain Index (I-HCPI). One hundred twenty-seven Italian-speaking owners of painful (i.e., osteoarthritic) and pain-free dogs were asked to complete the I-HCPI. The analyses performed confirmed that the I-HCPI clearly discriminated between healthy and painful dogs, as well as among different pain severity classes as evaluated by the veterinary surgeon or by using another scoring tool for chronic pain. Moreover, the I-HCPI showed the ability to detect pain severity decrease after pain treatment. In conclusion, the study confirmed that the I-HCPI is a new, reliable, and accurate tool to be used by Italian-speaking veterinarians and owners in order to measure and monitor chronic pain in dogs with osteoarthritis.

**Abstract:**

Pain assessment is of paramount importance for properly managing dogs with osteoarthritis (OA) pain. The aim of the present study was to develop and psychometrically validate the Italian version of the Helsinki Chronic Pain Index (I-HCPI). Owners of OA painful (*n* = 87) and healthy dogs (*n* = 40) were administered the I-HCPI once or twice after an eight-week meloxicam treatment. Sixty-nine owners of healthy and OA dogs also completed the Italian version of the Canine Brief Pain Inventory (I-CBPI). Pain on palpation on a 0–4 scale was assessed on all recruited dogs. Construct validity was tested both with hypothesis testing and principal component analysis, confirming the I-HCPI accurately measured chronic pain. Good convergent and criterion validity were shown through correlations with I-CBPI subscores and distribution among pain on palpation scores (*p* < 0.0001). The significant difference between the pre- and post-treatment I-HCPI scores (*p* < 0.0001) and Cohen’s effect size (2.27) indicated excellent responsiveness. The I-HCPI was shown to be reliable through communalities (range 0.47–0.90) and Cronbach α (≥0.95). Discriminative ability and cut-off point, as tested through Receiver Operating Characteristic analysis, showed excellent diagnostic accuracy with a threshold value of 11 (specificity 0.98 and sensitivity 0.94). The I-HCPI was confirmed to be a valid, sensitive, reliable, and accurate tool to discriminate between dogs with and without pain.

## 1. Introduction

Osteoarthritis (OA), otherwise referred to as osteoarthrosis or degenerative joint disease, represents one of the most frequent painful conditions in the canine population of all classes of age [1,2], with pain affecting at least 20% of the dog population older than 1 year in the United States [3]. In a recent study, 39.8% of recruited dogs had radiographic OA in at least one joint, and 23.6% had “clinical OA”, defined as the overlap of radiographic OA and joint pain in the same joint [4].

Without a doubt, pain related to OA is the quintessential example of chronic pain and is a two-component experience, with a sensory dimension (leading to lameness, stiffness, and movement reduction), and an emotional colouration, i.e., loss of interest in the environment, decreased social interactions, and reduced enjoyment of life. Both components significantly and gradually alter the animal behaviour and contribute to worsening the dog’s quality of life. In addition to the discomfort itself, chronic pain may indeed be associated with—or even exacerbate—neuro-behavioural problems like aggression, house soiling, anxiety, and cognitive dysfunction in affected pets [5,6,7]. Recognizing and assessing pain in OA dogs is paramount for treating it accordingly and limiting structural and emotional deterioration of the overall health status. Unfortunately, OA pain is underdiagnosed and undertreated, especially in young dogs [4]. 

As recently established by the 2020 International Association for the Study of Pain (IASP), verbal description is only one of several behaviours used to express pain [8]. Therefore, although dogs cannot self-report their pain, they do experience it, with the troublesome issue that pain needs to be estimated by a proxy, usually the veterinarian, the owner, or the caregiver. Moreover, it should be remembered that pain is an abstract construct, and as such, it is not directly measurable [9]. Clinical Metrology Instruments (CMIs) are entering clinical practice as semi-objective tools for estimating pain intensity, monitoring disease progression, and measuring treatment efficacy. A CMI is a questionnaire comprising several items, scored according to the observer assessment. The individual scores are summarised and used to calculate an overall instrument score. To be judged suitable, such questionnaires must be submitted to a psychometric evaluation confirming their validity, responsiveness, and reliability in every language that they have been tested in [10]. Today, a growing number of validated owner-reported outcome measures for assessing OA pain in dogs are available and have been recently reviewed [11]. Among them, the Helsinki Chronic Pain Index (HCPI) is a freely available owner-completed questionnaire originally developed in 2003 [12], and further validated in 2009, that gathers information on pain interference with dog’s behavior and locomotion, thus addressing both the emotional and sensory components of pain [13]. It consists of 11 questions regarding the dog’s mood, vocalization, willingness to move (at walk, trot, and gallop), willingness to play and jump, ease to lie down, rise, move after a long rest, and move after a major activity. Owners are asked to mark on a 5-point descriptive scale the answer that best describes their dog’s condition. The answers are then tied to a value (0 to 4) and summed to give a total index, ranging from 0 to 44 [14]. The HCPI has been used in several studies to assess pain in OA dogs [15,16], monitor the effect of medical therapy [14,17,18,19,20,21,22,23,24,25,26,27,28,29,30,31,32], surgical intervention [33,34], or rehabilitation [35], and is being regarded as an external criterion for the validation of other CMIs [36]. 

To be used in another language, any CMI must be properly translated, and again psychometrically tested, to ensure that the meaning and intent of the original items are maintained and the scale remains relevant [37,38,39,40]. To the authors’ knowledge, the originally validated HCPI [12,13] has been linguistically validated and psychometrically tested only in Portuguese [41]. The present study aimed to develop an Italian version of the HCPI (I-HCPI) and test its psychometric performance, under the hypothesis that the I-HCPI would have similar psychometric properties (i.e., validity, responsiveness, and reliability) as the original validated version.

## 2. Materials and Methods

### 2.1. Translation and Back-Translation

One of the authors (GdR), a native Italian speaker fluent in English, translated into Italian the owner questionnaire HCPI-E2 (freely available for research at https://dspace.uevora.pt/rdpc/bitstream/10174/19611/1/HCPI_E2.pdf accessed on 14 December 2023). The translation was then reviewed for accuracy by three bilingual external reviewers. Subsequently, the Italian version of the scale was back-translated by a native English speaker who was fluent in Italian and blind to the original scale. Finally, the back-translated scale was compared to the original one, and semantic equivalence was guaranteed by small adjustments to the Italian version of the scale.

### 2.2. The Italian Helsinki Chronic Pain Index (I-HCPI)

Like the original HCPI, the Italian version consisted of 11 questions addressing the dog’s mood, vocalization, willingness to walk, trot, gallop, play and jump, ease to lie down, rise, move after a long rest, and move after a major activity. Owners answered each question on a 5-point descriptive scale. The single answers were then tied to a 0 to 4 value and summed to give a total index score from 0 to 44. The I-HCPI is available as Appendix A.

### 2.3. Animals and Study Design

The data used in this study are part of research evaluating the efficacy and tolerability of a therapeutic protocol for OA pain, which was approved by the Local Ethical Committee of the University Perugia, Italy (protocol n. 2018-13) [31]. 

The study included 87 dogs with history, clinical signs, and radiographic evidence consistent with appendicular OA, suffering from associated chronic pain (i.e., lasting longer than 3 months), and presenting at the veterinarians’ attention from January 2020 to December 2021. Dogs with confounding comorbidities (i.e., pain arising from pathologies other than osteoarthritis), concurrent disorders interfering with locomotion, physical activity, or quality of life, as well as dogs that were pregnant or lactating, were not included. Moreover, a cohort of 40 clinically healthy dogs (i.e., without OA or any other disorder associated with chronic pain, as determined on the basis of patient history and routine physical examination by the veterinarian) were also recruited. The sample size was chosen based on the subject-to-item ratio between 3.6:1 (for healthy dogs) and 7.9:1 (for OA dogs) [42]. Dogs were stratified to gather information from both classes, i.e., healthy and OA dogs.

The owners were all native Italian speakers. After providing informed consent to participate in the study, 92 dog owners (i.e., the owners of all healthy and 52 OA dogs) completed the I-HCPI once, while the remaining 35 owners of OA dogs completed it twice, i.e., before and after eight weeks of meloxicam (Metacam, Boehringer Ingelheim Vetmedica GmbH, Ingelheim/Rhein, Germany; 1.5 mg/mL oral suspension on a tapering regimen, i.e., progressive 25% decrease of the original 0.1 mg/kg/day dose, on a biweekly basis, provided the pain severity (i) decreased at least by 30% during the full-dose regimen and (ii) did not increase in the last 15 days [31]). All eligibility criteria and treatment protocol details relevant to this dog population are comprehensively described in reference [31] to which the reader is referred. Moreover, 69 owners (i.e., the owners of all healthy dogs and 29 OA dogs) also completed the validated Italian version of the Canine Brief Pain Inventory (I-CBPI) [43]. The I-CBPI consists of 11 questions, with four addressing pain severity (Pain Severity Score—PSS), six addressing pain interference with dog’s daily activities (Pain Interference Score—PIS), and the last one concerning the dog’s quality of life (QoL) [43].

Pain on palpation was assessed by veterinary surgeons with extensive experience in orthopaedics on all recruited dogs on a 5-point scale modified from previous studies [44]. The pain was scored as 0 = no sign of pain; 1 = mild pain (dog turns head in recognition); 2 = moderate pain (dog pulls limb away or wants to move away); 3 = severe pain (dog vocalizes or becomes aggressive); 4 = extreme pain (dog does not allow palpation). 

### 2.4. Psychometric Tests and Statistical Analysis

The *t*-test and the Fisher’s test were used to analyse the age and reproductive status distribution, respectively, of healthy and OA dogs. Psychometric tests were performed to assess construct and criterion validity, responsiveness, reliability (internal consistency), and cut-off point for the presence of chronic pain, as described below. The statistical analyses were performed using SAS software, version 9.4 (SAS Institute, Cary, NC, USA). The level of significance was set at *p* < 0.05.

#### 2.4.1. Validity

Construct validity is central to establishing the overall validity of a tool, and represents how well scores are consistent with the theoretical hypothesis, i.e., if they are indicative of the theoretical construct [45]. In terms of pain, for instance, the severity is expected to increase during chronic progressive disorders, like OA, while it is expected to decrease with analgesic treatment. In the present study, the I-HCPI construct validity was initially tested using the hypothesis-testing methodology, i.e., by assessing whether healthy dogs had lower scores compared to OA dogs. The Wilcoxon rank-sum test was used to compare the I-HCPI scores of independent samples (i.e., healthy vs. OA dogs). 

Construct validity of the I-HCPI was also assessed using factor analysis, which distinguishes the underlying dimensions (dimensionality), establishing the relationship between the instrument items (questions) and each dimension [10,46]. First, the Kaiser-Meyer-Olkin measure (KMO) [47] was used to verify the sampling adequacy for the analysis, with a KMO value >0.6 indicating that data were suitable for component analysis. Principal Component Analysis (PCA) was then performed with and without varimax rotation. Items with loading values ≥0.32 were retained in the overall I-HCPI interpretation. Only constructs with an eigenvalue >1 were retained for interpretation (i.e., Kaiser Criterion) [48,49]. Catell’s scree test (i.e., visual investigation of the significant break in the eigenvalue scree plot) was also used, in order to decide the components to be retained [49].

Convergent validity (i.e., the extent of correlation between two measures of a construct that theoretically should be related) [50] was tested under the hypothesis that the I-HCPI score negatively correlated to QoL on I-CBPI. Spearman’s correlation was used, with absolute rho (ρ) values being interpreted as follows: ρ ≤ 0.35 poor or no correlation; 0.35 < ρ < 0.68, moderate correlation; ρ ≥ 0.68 strong correlation; ρ ≥ 0.90 very high correlation [51]. 

Criterion validity (which is an estimate of the extent to which a measure agrees with a gold standard) [52] was tested, both against veterinarians’ ratings of pain on palpation (0–4 scale) and the two components of the I-CBPI (i.e., PSS and PIS). The Kruskal—Wallis test and Pearson’s correlation were used, respectively, with Pearson’s correlation coefficient (r) being considered to indicate a strong correlation for values higher than 0.68.

#### 2.4.2. Responsiveness

Responsiveness or sensitivity to change assesses the ability of a tool to detect significant changes in the expected direction [53]. Hypothesis testing was used to assess whether OA dogs had lower scores after being treated with meloxicam. The Wilcoxon signed-rank test was used to compare the I-HCPI scores of related samples (i.e., pre- vs. post-treatment scores). The difference between pre- and post-treatment I-HCPI scores was also expressed with Cohen’s *d* effect size (difference between pre- and post-treatment mean scores divided by the pooled standard deviation, 0.2 being considered small, 0.5 as medium, and 0.8 or greater as large [54]).

#### 2.4.3. Reliability

Reliability assesses whether the instrument is measuring something in a reproducible and stable way, and was tested through the internal consistency method, which estimates the correlation between the items of the I-HCPI [10]. In particular, communalities (i.e., proportions of variance for each item that can be explained by the component) were evaluated, with values >0.40 indicating the item is related to the other items [42]. Moreover, item-total correlations (i.e., correlations between individual items and the I-HCPI total score when that item is omitted) were calculated, with values >0.2 being retained as previously suggested [13]. Finally, Cronbach α value was calculated in order to measure the extent to which the item responses are correlated to each other; α values >0.7 were considered to be indicative of a higher internal consistency [10]. 

#### 2.4.4. Diagnostic Accuracy and Cut-Off Point 

The I-HCPI score corresponding to pain presence was identified through the Receiver Operating Characteristic (ROC) analysis [55]. The resulting ROC curve plotted true positive (sensitivity) against false positive rates (1—specificity) across all possible cut-points. The area under the ROC curve (AUC) represented the global discriminative ability of the I-HCPI (i.e., the accuracy of the instrument in discerning between dogs with and without pain) [56]. The greater the AUC value, the higher the discriminatory ability (1.0 = perfect accuracy) [57]. The Youden index, which provides the optimal diagnostic cut-point, was additionally calculated. The maximum value of the index was used as a criterion for selecting the optimum cut-off point. The accuracy index (i.e., a measure of the discriminative ability of the cut-off) was finally calculated with the following formula: “N. of subjects correctly classified/total N. of subjects”. An accuracy index equal to 1 would indicate that there is a threshold value that perfectly separates dogs with and without pain. 

## 3. Results

### 3.1. Demographics

A total of 127 dogs were evaluated. The mean age was 4.8 ± 3.3 and 9.0 ± 3.6 years and the mean body weight was 21.6 ± 10.8 and 31.0 ± 11.6 kg for healthy and OA dogs, respectively (Table 1). *t*-test for unequal variances showed that healthy dogs were on average younger and lighter than OA dogs (*p* < 0.0001 for both comparisons). As summarized in Table 2, male and female dogs were equally distributed in the healthy group (20 each). On the contrary, males slightly outnumbered females in the OA group (46 vs. 41). Neutered dogs were significantly more represented in the OA compared to the healthy group (*p* < 0.0288 at Fisher’s test, Table 2). 

Thirty-three different breeds were represented, with mixed breeds being the most common (18/87 and 14/40 in OA and healthy dogs, respectively), followed by Labrador Retriever, German Shepherd, and Golden Retriever (Figure 1). The less represented breeds (i.e., those with 1 dog each) are not shown, and overall included 9 OA and 12 healthy dogs. 

### 3.2. Construct Validity

Healthy dogs showed significantly lower scores compared to OA dogs (*p* < 0.0001), thus confirming the construct validity of the I-HCPI (Figure 2; Table 3).

Construct validity was also assessed using PCA. The KMO test (KMO = 0.94) showed common factors which were suitable for performing PCA. The PCA resulted in the extraction of two components with an eigenvalue >1 (8.02 and 1.03, respectively) (Figure 3), explaining 72.9% and 9.4% of the variance respectively, with a total cumulative variance of 82.3%.

At varimax rotation, the first component exhibited the highest loadings on items related to dog’s locomotion (i.e., I-HCPI items 4 to 11), with loading values ranging from 0.80 to 0.86, while item 3 (i.e., vocalization) showed a somewhat lower loading (loading value: 0.65). The second component loaded on items relating to the dog’s mood and playfulness (i.e., I-HCPI items 1 and 2, with loading values of 0.87 and 0.83, respectively). Data are summarized in Table 4. 

Based on these data, and the visual analysis of the scree plot (Figure 3), we considered the I-HCPI to be best explained as a single component index, referred to as chronic pain. The decision was made by combining the Cattell’s Scree test result with the Kaiser Criterion result [49], with the choice being further supported by good internal consistency, as will be described in the Reliability paragraph. Table 5 shows the loadings from the one-component solution. Loading values ≥0.64 were observed for all items but one (i.e., item 3, vocalization), which, however, exhibited a loading of 0.43, well above the acceptable threshold value (=0.32) [48]. 

### 3.3. Convergent Validity

There was a strong, significant negative correlation between the I-HCPI and QoL scores on I-CBPI (ρ = −0.82, *p* < 0.0001), confirming that increasing pain was associated with decreasing QoL. 

### 3.4. Criterion Validity

As illustrated in Figure 4, the I-HCPI scores were significantly distributed among 0–4 pain on palpation scores (*p* < 0.0001). 

Moreover, a strong and significant correlation with both pain severity and pain intensity subscores of the I-CBPI was found (r = 0.84; *p* < 0.0001, both for PSS and PIS), further supporting criterion validity (Figure 5).

### 3.5. Responsiveness

A significant difference was found between pre- and post-treatment I-HCPI scores (*p* < 0.0001) (Figure 6), with the effect size being 12.2 (95% CI 9.5; 14.9), and Choen’s *d* = 2.27.

### 3.6. Reliability

For all questions, communality ranged from 0.47 (vocalization only) to 0.90 (Table 6), being well above the 0.40 cut-off value, thus highlighting strong internal consistency. Item-total correlations were all >0.20. The Cronbach’s α coefficient was ≥0.95 (Table 6).

### 3.7. Diagnostic Accuracy and Cut-Off Point

At ROC analysis, the I-HCPI showed excellent diagnostic accuracy, given an AUC equal to 0.99 (Figure 7). The maximum value of the Youden index (i.e., 0.92) corresponded to the I-HCPI score of 11, which was thus selected as the optimum cut-off to discriminate the presence of chronic pain. The estimates and 95% confidence intervals (CI) for specificity and sensitivity of the I-HCPI, using the above threshold value as an indicator of chronic pain, were 0.98 (95% CI: 0.93; 1.00) and 0.94 (95% CI: 0.89; 0.99), respectively. The I-HCPI score ≤ 11 allowed to correctly classify as “pain free” 39 out of the 40 healthy dogs, while the I-HCPI score > 11 correctly classified as “in pain” 82 out of 87 OA painful dogs. The accuracy index was 0.95.

## 4. Discussion

Measuring pain in a valid and reliable way is of crucial importance for its proper management, and even more so when one deals with patients that are unable to self-report, like animals [9]. Thereby, easy to use and accurate tools for measuring OA pain may be of great help in the veterinary clinical practice. 

Based on data collected from 87 OA and 40 healthy dogs, here we performed a psychometric validation of the Italian translation of the HCPI. Healthy and OA dogs significantly differed by age and reproductive status. Essentially, older, heavier, and neutered dogs were more likely to be affected by OA-associated pain. This is in accordance with the results of a recently published systematic review, highlighting body weight, neuter status, and age as important risk factors for OA [3]. Although on the basis of the information collected in the present study it could not be determined if body weight actually reflected body condition (i.e., lean/overweight/obese) rather than breed size, the link between body weight and OA apparently agreed with previously reported data, with an increasing proportion of dogs with OA being also overweight or obese [58].

The I-HCPI was here shown to accurately measure what it is supposed to, i.e., chronic pain. This emerged from the study of construct validity, which was here assessed both with a hypothesis testing procedure and with PCA. At hypothesis testing, the I-HCPI was shown to efficiently reveal pain, as healthy dogs exhibited significantly lower scores (range 0–11) compared to OA dogs (range 6–39), in accordance with the original investigation [12]. The PCA results were very similar to those obtained in two previous studies [13,36]. Indeed, two factors with eigenvalues >1 were extracted before varimax rotation, while after rotation two components were identified, with a third component (i.e., vocalization) being potentially present, given the lower loading values. Based on the same considerations made by previous validation studies [13,41], we considered the I-HCPI to be best explained as a single component index, and all the items but one (vocalization) showed high loading values ranging from 0.64 to 0.95 (Table 4). This indeed indicated that each of the 11 items strongly related to a single component (i.e., chronic pain), and confirmed the unidimensionality of the I-HCPI. The finding relative to the lower loadings of vocalization has already been reported and addressed [12], with the reason being possibly linked to the difference between the audible complaint of a dog in chronic pain (e.g., whimpers) and what the owner considers to be vocalization (i.e., crying out in pain). 

Good convergent and criterion validity of the I-HCPI were observed in the present study. To the best of our knowledge, this is the first time that convergent validity has been tested for HCPI, with the negative correlation with QoL being significant and strong. It is moreover worth noting that here the criterion validity of the I-HCPI was successfully tested both against pain on palpation (i.e., a veterinarian-assessed measure) and the pain subscores (severity and interference) of the owner-reported CBPI. The results are in accordance with previous studies [13,36], while the Portuguese validation observed only a moderate correlation, which may depend on the smaller sample or the continuous nature of the scale being correlated (i.e., VAS) [41]. 

The significant difference between the pre- and post-treatment scores, together with the large effect size (Cohen’s *d* = 2.27), supported the excellent responsiveness (i.e., sensitivity to changes) of the I-HCPI in accordance with the original validation study [13]. On the contrary, responsiveness was not specifically addressed in the study by Walton and colleagues (2013) [36], and could not be demonstrated in the Portuguese version, since no difference emerged between carprofen and placebo groups [41]. The smaller sample of studied dogs, the different pain-reliever (meloxicam vs. carprofen), or the multiple time points considered in the study by Matsubara and colleagues (2019) [41] may account for the discrepancy. 

The present study showed that the I-HCPI measures chronic pain in a reproducible and stable way, as tested by estimating the correlation between the 11 items constituting it (i.e., internal consistency). The Cronbach’s α coefficient was well above the acceptable level, as were the communalities and interitem correlations. The reliability of the I-HCPI was thus confirmed, in agreement with the original validation of this CMI [13]. 

Finally, for the first time, the discriminative ability of the scale was assessed and turned out to be excellent. In particular, the ROC curve AUC was found to be 0.99. Given that the score for perfect accuracy is 1, the analysis confirmed that the I-HCPI has excellent accuracy in classifying subjects with and without pain. This was also supported by the high sensitivity and specificity values. The ROC analysis also allowed for selecting the optimal cut-off point, i.e., the I-HCPI value able to separate dogs with and without pain. The cut-off point was found to be 11, corresponding to the maximum value of the Youden index. The accuracy in discriminating dogs with and without pain was considered excellent (0.95 out of 1). The finding completely agreed with the suggestion of Hielm-Björkman and colleagues (2003), who proposed 11 to be the maximum HCPI score for healthy dogs, and 12 the minimum index for painful ones [12]. 

## 5. Conclusions

The psychometric testing of the I-HCPI confirmed that it is a valid, sensitive, reliable, and accurate CMI for assessing chronic pain in dogs affected by OA. Moreover, the ROC-based methodology allowed us to establish, for the first time, the I-HCPI score of 11 as an accurate diagnostic cut-point to discriminate between dogs with and without pain. The present findings provide Italian-speaking clinicians and researchers with a useful metrology instrument to evaluate the severity of chronic pain in OA dogs, as well as monitor the animal’s response to treatment interventions in terms of pain relief.

## Figures and Tables

**Figure 1 animals-14-00083-f001:**
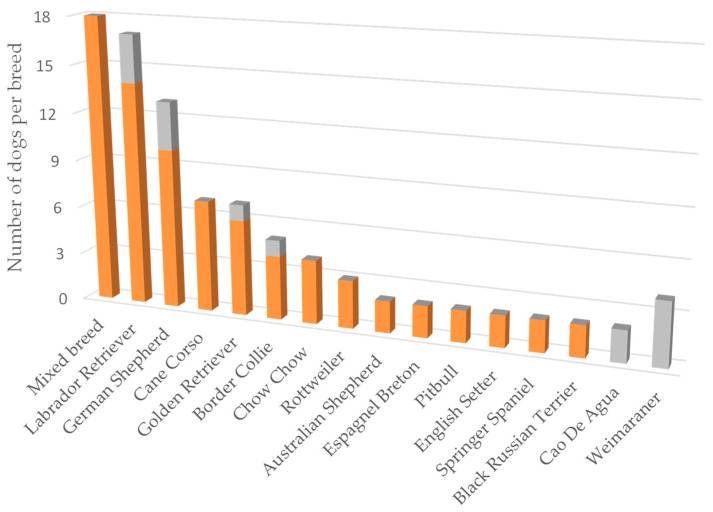
The most represented breeds in the OA (orange) and healthy (grey) dogs. Only breeds represented by more than one dog per group are illustrated.

**Figure 2 animals-14-00083-f002:**
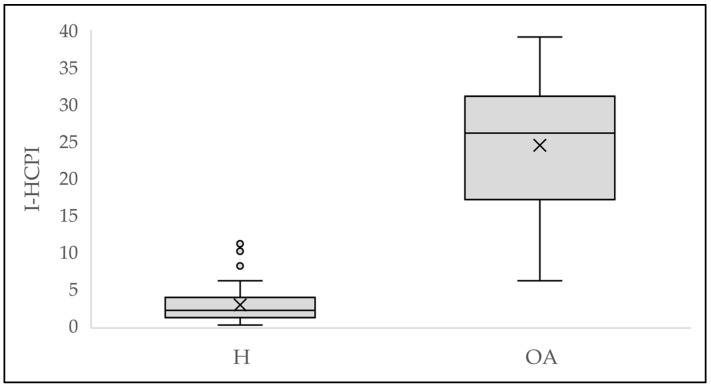
Difference in I-HCPI scores between healthy (H) and osteoarthritis (OA) dogs. The crosses represent the mean; the horizontal line in the rectangle represents the median; the dots represent the outlier values.

**Figure 3 animals-14-00083-f003:**
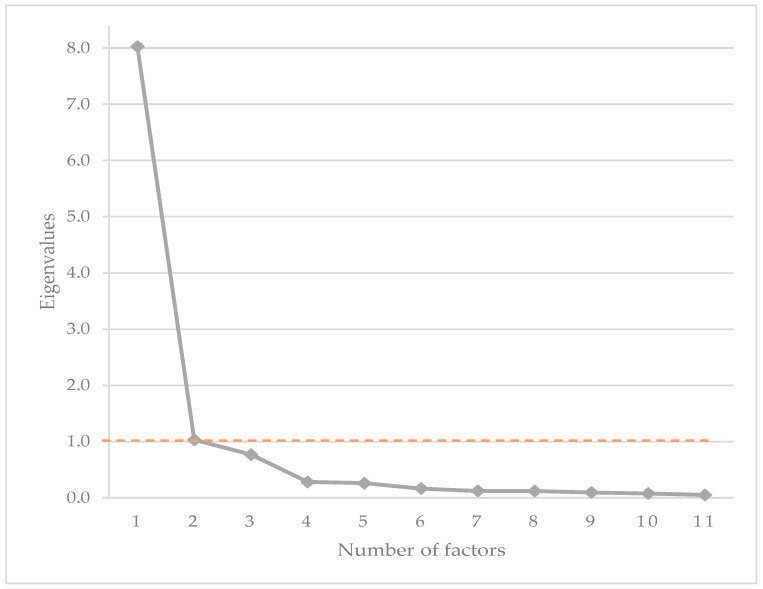
Scree plot of the eigenvalues of the I-HCPI. The orange dashed line represents the cut-off score for the eigenvalue (i.e., 1 according to Kaiser criterion).

**Figure 4 animals-14-00083-f004:**
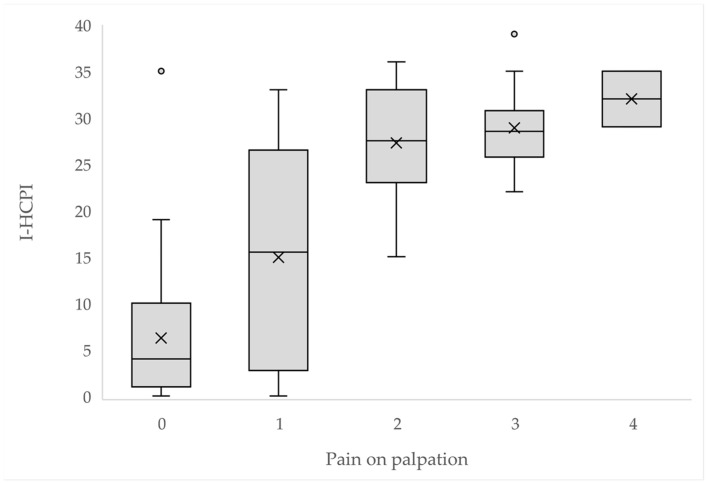
I-HCPI score distribution by pain on palpation. The crosses represent the mean; the horizontal line in the rectangle represents the median; the dots represent the outlier values.

**Figure 5 animals-14-00083-f005:**
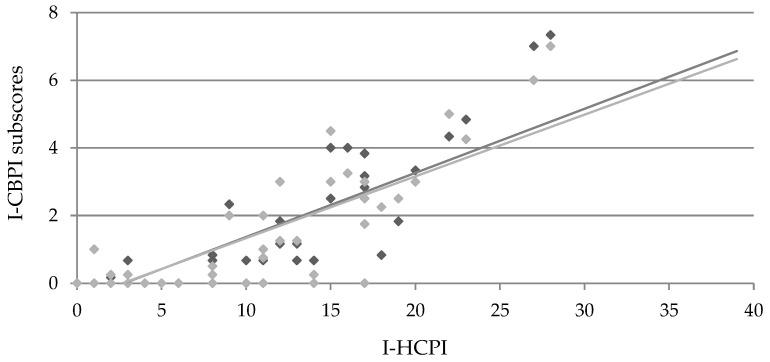
Pearson’s correlation of I-HCPI and I-CBPI subscores (i.e., pain severity, light grey; pain interference, dark grey). Coefficient r = 0.84 and *p* < 0.0001 for both correlations.

**Figure 6 animals-14-00083-f006:**
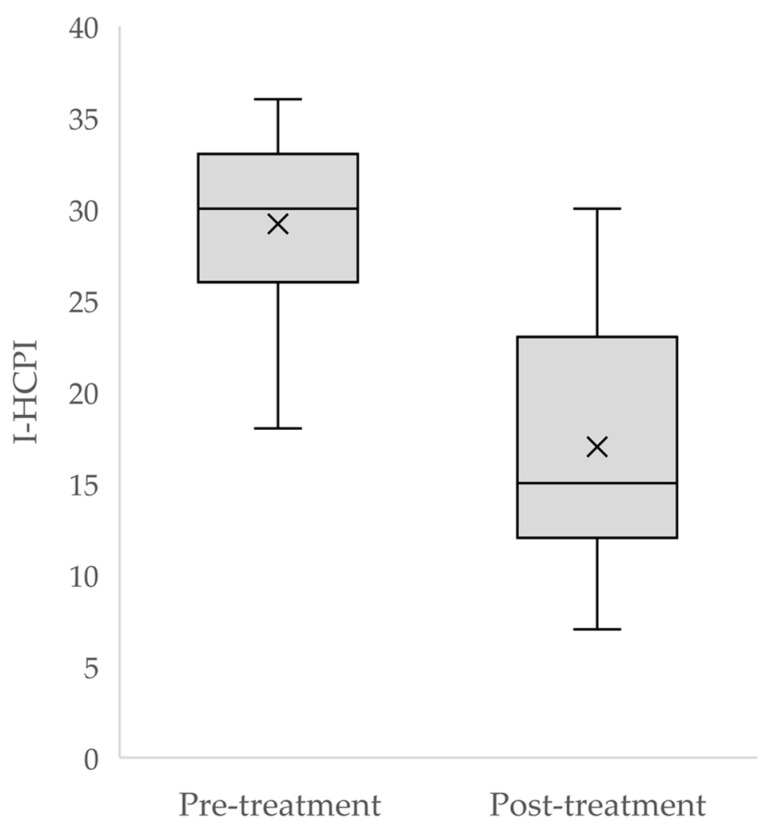
Distribution of I-HCPI scores before and after treatment. The crosses represent the mean, while the horizontal line in the rectangle represents the median.

**Figure 7 animals-14-00083-f007:**
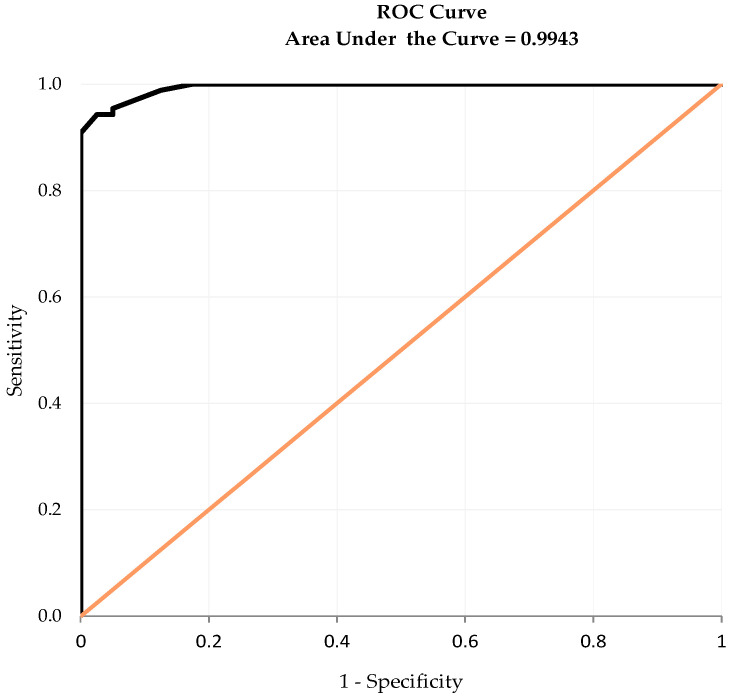
Receiver Operating Characteristic (ROC) curve.

**Table 1 animals-14-00083-t001:** Distribution according to age and body weight in osteoarthritis (OA), healthy and all dogs.

	Age (years)			Body Weight (kg)		
	OA (N = 87)	Healthy (N = 40)	All (N = 127)	OA (N = 87)	Healthy (N = 40)	All (N = 127)
Min	0.5	1.2	0.5	8.3	8	8
Q1	6.8	1.7	4.0	22.9	12	19.8
Median	9.0	4.0	8.0	32.8	20.5	28
Q3	11.5	6.9	11.0	38.7	28	37
Max	16.0	14.5	16.0	60	52	60
Mean	9.0	4.8	7.7	31.0	21.6	28.1
Std	3.6	3.3	4.0	11.6	10.8	12.1

**Table 2 animals-14-00083-t002:** Distribution according to sex and reproductive status in osteoarthritis (OA), healthy and all dogs.

	OA		Healthy		All	
	N	%	N	%	N	%
Fe	6	6.9	10	25.0	16	12.6
Fs	35	40.2	10	25.0	45	35.4
Me	35	40.2	17	42.5	52	40.9
Mn	11	12.6	3	7.5	14	11.0
Total	87	100	40	100	127	100

Fe: Female Entire; Fs: Female Spayed; Me: Male Entire; Mn: Male Neutered.

**Table 3 animals-14-00083-t003:** I-HCPI scores in healthy and osteoarthritic (OA) dogs.

	OA	Healthy	All
N	87	40	127
Min	6	0	0
Q1	17	1	5
Median	26	2	18
Q3	31	3.5	28
Max	39	11	39
Mean	24.4	2.7	17.5
StdErr	0.88	0.47	1.09

**Table 4 animals-14-00083-t004:** PCA loading values for I-HCPI items after varimax rotation of the two eigenvalues >1.

		PCA Loading	
Item	Description	Factor 1	Factor 2
I-HCPI01	Mood	0.20	0.87
I-HCPI02	Play	0.28	0.83
I-HCPI03	Vocalization	0.65	−0.21
I-HCPI04	Walking	0.82	0.45
I-HCPI05	Trotting	0.83	0.45
I-HCPI06	Galloping	0.80	0.48
I-HCPI07	Jumping	0.80	0.50
I-HCPI08	Lying down	0.83	0.37
I-HCPI09	Getting up	0.86	0.38
I-HCPI10	Movement after rest	0.84	0.44
I-HCPI11	Movement after major exercise	0.82	0.46

**Table 5 animals-14-00083-t005:** I-HCPI values (mean and SD) for items totalled and individual items with the corresponding PCA loading for the one-component solution.

Item	Description	Mean	SD	PCA Loading
I-HCPI	Items totaled	17.54	12.28	NA
I-HCPI01	Mood	1.26	1.03	0.64
I-HCPI02	Play	1.14	1.04	0.68
I-HCPI03	Vocalization	0.69	0.89	0.43
I-HCPI04	Walking	1.51	1.28	0.93
I-HCPI05	Trotting	1.86	1.51	0.95
I-HCPI06	Galloping	1.95	1.46	0.93
I-HCPI07	Jumping	2.10	1.57	0.95
I-HCPI08	Lying down	1.39	1.24	0.89
I-HCPI09	Getting up	1.78	1.42	0.92
I-HCPI10	Movement after rest	1.87	1.38	0.95
I-HCPI11	Movement after major exercise	1.99	1.41	0.94

**Table 6 animals-14-00083-t006:** Communalities, Item-total correlations, and Cronbach’s α I-HCPI for the 11 I-HCPI items.

Item	Description	Communalities	Item-Total Correlations	Cronbach α
I-HCPI	Items totaled	NA	NA	0.96
I-HCPI01	Mood	0.80	0.64	0.96
I-HCPI02	Play	0.77	0.68	0.96
I-HCPI03	Vocalization	0.47	0.45	0.97
I-HCPI04	Walking	0.87	0.93	0.96
I-HCPI05	Trotting	0.90	0.95	0.95
I-HCPI06	Galloping	0.87	0.93	0.96
I-HCPI07	Jumping	0.90	0.95	0.95
I-HCPI08	Lying down	0.82	0.89	0.96
I-HCPI09	Getting up	0.88	0.92	0.96
I-HCPI10	Movement after rest	0.90	0.95	0.95
I-HCPI11	Movement after major exercise	0.88	0.94	0.95

## Data Availability

Data is contained within the article and Appendix A.

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
