# Peer review of "Psychometric Testing and Validation of the Italian Version of the Helsinki Chronic Pain Index (I-HCPI) in Dogs with Pain Related to Osteoarthritis"

_animals, 2023, doi:10.3390/ani14010083_

Round 1
Reviewer 1 Report
Comments and Suggestions for Authors
Interesting and valuable study – nicely reported. I just have a few minor comments.
Simple Summary
Not sure if it is as simple as it could be – still includes quite a lot of technical language.
Abstract
Could include p-value in abstract for significant difference in the pre/post treatment scores.
Introduction
Line 59 – ‘is the quintessential of chronic pain’ not sure this statement makes grammatical sense.
Line 99 - I am just wondering in terms of rationale – how does changing the language potentially affect the properties of a CMI? You do briefly mention that they need to be properly tested and translated – I think this aspect should be further emphasised as this really the main rationale for the need for the present study as I understand it. Have other studies looked at the impact of translating other CMI?
Methods
Line 140 – where it mentions that dogs in the OA group were measured pre/post eight weeks of meloxicam – I assume this prescription was from the dogs’ veterinarian? I also assume no dogs were withheld recommended analgesics for the purpose of the study? (e.g. to get the ‘before’ measure).
Line 167 – why was the Wilcoxon used to compare the healthy vs OA dogs (independent samples) – this test is the non-parametric version of the paired t-test so would normally be used on related samples? You also use the Wilcoxon (correctly) for related samples in line 193 (pre/post treatment).
Results
Line 234 – Think German Shepherd might be spelled incorrectly.
Figure 3 – could explain what the orange dashed line represents in the caption.
Discussion
Good discussion – no comments.
Reviewer 2 Report
Comments and Suggestions for Authors
The aim of the authors of the submitted article is to validate the translation into Italian of the Helsinki Chronic Pain Index.
The HCPI is one of the most popular chronic pain assessment scales available in English. As the authors point out, a Portuguese translation is also available. Pain assessment scales are completed by pet owners, so it is important that they are easy to understand. Misunderstanding of English-language terms or incorrect translations can cause errors and affect the reliability of the assessment. Therefore, it seems advisable to use the scale in the native language of the pet owner. Validation of the scale was based on the assessment of chronic pain in OA. The choice of disease is appropriate as it affects a large number of animals and is chronic in nature. Animals do not need to be hospitalised in the course of the disease, and pain relief schemes in OA are known.
The authors first attempted a multiple translation of the available scale in order to get the results as close as possible to the available English scale. The resulting document was implemented in clinical practice and made available to 137 dog owners ( 87 dogs with OA and 40 healthy dogs).
In the introduction to the article, the authors briefly describe the characteristics of chronic pain in OA and refer to the principles of the original version of the HCPI. The introduction is concise, easy to understand and well written. In the Materials and methods section, the authors accurately describe the course of the experiment, including the inclusion and exclusion criteria. In my opinion, information on the dose of meloxicam used per kg body weight and the route of administration was lacking. It would also be useful to include information on whether the animals in the patient group had previously received other treatments (meloxicam, laser therapy, cryotherapy, etc.) and to discuss whether early treatment could have affected the results of the experiment. Were any additional recommendations given to the owners during the evaluation period (restriction of exercise, change in diet, etc.)? In the Materials and methods, the statistical analysis is described in a very extensive way, divided into validity, responsiveness, reliability and accuracy. The results are presented in the form of clear tables and graphs. The discussion is well written and includes interesting references to the Portuguese translation of the scale. The conclusions correspond with the stated aim of the study and with the results obtained.
The supplementary files containing the document that the owners filled in is an important part of the article.
The article contains 55 actual and references , but there are many self-citations (7).
In summary, the article is very interesting and, in my opinion, has potentially high clinical value. It has some limitations ( the Italian version of the HCPI narrows the audience), but may inspire translation of the scale into other languages. I recommend the article for publication after minor corrections pointed out above in the review.
Reviewer 3 Report
Comments and Suggestions for Authors
“Psychometric testing and validation of the Italian version of the 2 Helsinki Chronic Pain Index (I-HCPI) in dogs with pain related 3 to osteoarthritis.”
Dear Authors,
Congratulations on your article. As the authors said, having the ability to objectively assess the pain level of a veterinary patient is crucial. The article is interesting and well written, I have only a few observations:
Line 99: why was the Finnish version not considered? “Psychometric testing of the Helsinki chronic painindex by completion of a questionnaire
in Finnish by owners of dogs with chronic signs
of pain caused by osteoarthritis”, Anna K. Hielm-Björkman, DVM, PhD; Hannu Rita, PhD; Riitta-Mari Tulamo, DVM, PhD, 2009
Line 133: “whitout OA…” How did you determine this?
Line 140: Which protocol did you use? What is the name of the meloxicam used and at what concentration?
Line 146: what was the level of experience of the veterinary surgeons? Could authors add this information?
Line 150: Have you adopted a strategy to differentiate the behavioural intolerance response from the pain intolerance response? If yes, it should be described
Reviewer 4 Report
Comments and Suggestions for Authors
The paper contains nothing new, not even the claim of being the first investigators testing and evaluating an Italian language version of the Helsinki chronic pain index is new. A group lead by researchers of the University of Turin (Italy) published already in 2018 a much similar, if not the same study:
Elisa, Martello, et al. "Evaluation of the efficacy of a dietary supplement in alleviating symptoms in dogs with osteoarthritis." JOURNAL OF FOOD AND NUTRITION 4 (2018): 1-8.
In this study, chronic pain was assessed with an Italian adapted version of the validated Helsinki Chronic Pain Index (HCPI).
This above paper, however, is not cited in this manuscript under review; to give the benefit of doubt, this omission of the citation happened likely by error, since both papers on the same topic originate from Italian Universities.
My recommendation would have been to reject the manuscript because of overall lack of novelty, and because of non-citation and non-recognition of a previous study on the same subject using the same methods and arriving at similar results.
.
Round 2
Reviewer 4 Report
Comments and Suggestions for Authors
suggest acceptance with minor revisions, please see comments in attached Word file

Author Response
Please, see the attached file.
